# Development of a Bitterness Sensor Using Partially Dissociated Amine Compounds

**DOI:** 10.3390/s24175553

**Published:** 2024-08-28

**Authors:** Yuyang Guo, Xiao Wu, Hidekazu Ikezaki, Kiyoshi Toko

**Affiliations:** 1Graduate School of Information Science and Electrical Engineering, Kyushu University, 744 Motooka, Nishi-ku, Fukuoka 819-0395, Japan; guo.yuyang.012@s.kyushu-u.ac.jp; 2Department of Information Electronics, Faculty of Engineering, Fukuoka Institute of Technology, 3-30-1 Wajiro-higashi, Higashi-ku, Fukuoka 811-0295, Japan; 3Intelligent Sensor Technology, Inc., 5-1-1 Onna, Atsugi-shi 243-0032, Japan; ikezaki.hidekazu@insent.co.jp; 4Research and Development Center for Five-Sense Devices, Kyushu University, 744 Motooka, Nishi-ku, Fukuoka 819-0395, Japan; 5Food and Health Innovation Center, Nakamura Gakuen University, 5-7-1 Befu, Jonan-ku, Fukuoka 814-0198, Japan; 6Institute for Advanced Study, Kyushu University, 744 Motooka, Nishi-ku, Fukuoka 819-0395, Japan; 7Graduate School of Nutritional Sciences, Nakamura Gakuen University, 5-7-1 Befu, Jonan-ku, Fukuoka 814-0198, Japan

**Keywords:** electronic tongue, lipid polymer membrane, bitterness evaluation, taste sensor, potentiometry

## Abstract

This study focused on developing an advanced bitterness sensor designed to minimize interference from common anions such as nitrate (NO_3_^−^) and iodide (I^−^) by incorporating partially dissociated amine compounds into the sensor membrane. The conventional bitter sensor (C00) uses fully dissociated quaternary ammonium salt tetradecyl ammonium bromide (TDAB), which typically exhibits high responses to these anions, leading to inaccurate bitterness assessments. To address this issue, we explored the use of three partially dissociated amines—oleylamine (OAm), dioctadecylamine (DODA), and tridodecylamine (TDA)—as lipids in the membrane components. We fabricated sensor membranes and tested their ion selectivity, interference resistance to anion, and sensitivity to iso-alpha acids (IAAs), representative bitter compounds in beer. The results showed that the membranes with partially dissociated amines significantly reduced anion interference. Notably, the sensitivity of the TDA membrane to IAAs was 80.4 mV/dec in concentration, exceeding the 68.5 mV/dec of the TDAB membrane. This enhanced sensitivity, coupled with reduced anion interference, reveals a novel property of partially dissociated lipids in taste sensors, distinguishing them from fully dissociated lipids. These findings pave the way for the development of sensors that can accurately assess a bitter taste and have potential applications in the food and beverage industry.

## 1. Introduction

Bitterness evaluation is particularly important in the development and marketing of food and pharmaceutical products. It aids in balancing flavors and developing products that meet consumer preferences [1,2,3,4]. Additionally, bitterness can serve as an indicator of the quality and safety of food products [5,6,7] and helps determine whether certain compounds may be harmful to the human body in practical bitterness evaluation [8,9].

Taste sensors, also known as electronic tongues, have been developed to numerically detect the flavors of food. Different electronic tongues utilize various sensing methodologies for the measurement of food flavors. Some of them employ electrochemical methods, including potentiometry [10,11,12,13,14] or voltammetry [15,16,17,18,19], as well as biomimetic biosensing [20,21], optical methods [22], and others. Toko et al. proposed and developed a type of potentiometric taste sensing system based on sensor electrodes equipped with lipid polymer membranes as the reception units [23]. The lipid polymer membranes mimic the structure and function of the lipid bilayer of human gustatory cells in which membrane proteins are embedded. The sensor electrodes can specifically mimic five basic tastes of human tongue: saltiness, sourness, bitterness, sweetness, and umami, along with a broad sensor of taste, astringency. The sensor electrode’s selective response to taste quality, termed *global selectivity*, differs from the “one-to-one” correspondence to specific chemical substances seen in biosensors like enzyme biosensors. It interacts with taste substances through electrostatic and hydrophobic interactions, leading to variations in the membrane surface charge density. These changes are detected through membrane potential and used as the outputs, allowing the numerical measurement of food flavors.

The sensor electrode C00, equipped with a lipid polymer membrane composed of positively charged lipid tetradodecylammonium bromide (TDAB), plasticizer 2-nitrophenyl octyl ether (NPOE), and support material polyvinyl chloride (PVC), is used to quantify the bitterness of acidic bitter substances present in food and beverages, such as the bitter compounds contained in coffee or beer [24,25,26]. The representative bitter substance in beer is known as iso-α acids (IAAs), commonly found in hops, which are one of the key ingredients in beer [27,28]. IAAs release hydrogen ions, and the main body of IAAs carry a negative charge and exhibits hydrophobicity after dissociation in water. Therefore, IAAs can interact with the lipid polymer membrane of C00 through electrostatic and hydrophobic interactions, thereby causing changes in the membrane potential.

However, some limitations of C00 have been identified in practice use [29]. When evaluating the taste of foods rich in iodide ions (I^−^), such as seaweed and kelp, and those containing nitrate ions (NO_3_^−^), like green vegetables, these foods, which do not possess a strong bitter taste, showed high sensor responses. According to Standard Tables of Food Composition in Japan, the concentration of I^−^ in kelp can reach up to approximately 20 mM, while the concentration of I^−^ in kelp soup is about 0.2–1 mM. The concentration of NO_3_^−^ in green vegetables such as komatsuna (Japanese mustard spinach) can reach up to 80 mM, and after diluted for testing, the detection results are about 16–20 mM [30]. On the other hand, sensory tests have shown that NO_3_^−^ at concentrations around 30 mM does not exhibit a particularly strong bitterness [31]. The discrepancy makes it difficult for C00 to accurately evaluate the bitterness of such foods containing NO_3_^−^ and I^−^.

In our previous study, it was reported that replacing fully dissociated lipids with partially dissociated lipids in the membrane changes the electrical properties of the membrane [32]. The study analyzed two types of negatively charged lipids in sensor membranes, phosphoric acid di-n-decyl ester (PADE) and tetrakis[3,5-bis (trifluoromethyl)phenyl]borate (TFPB), to test the differences in response to cations. PADE is a partially dissociated lipid, while TFPB is fully dissociated. When the test solutions contain the same concentrations of cations such as Li^+^, Na^+^, K^+^, and Cs^+^, the PADE membrane exhibits a lower response to all solutions and no longer demonstrates significant anion selectivity compared to the TFPB membrane. PADE mitigates the effect of the Hofmeister series on cations.

Inspired by the research on PADE and TFPB, we considered that the problem observed with the sensor electrode C00 is related to the lipid component, tetradodecylammonium bromide (TDAB) in the lipid polymer membrane. TDAB is a quaternary ammonium salt, also known as QAS. As a QAS, the dissociation of TDAB is complete in aqueous solutions. When used as membrane lipid and dissociated in solution, bromide ions are released from TDAB, rendering the membrane surface to be positively charged. As a result, various types of anions can be electrostatically adsorbed to the TDAB membrane and cause a potential change, leading to interference in bitterness assessment. To enhance the selectivity of the bitterness sensor, it is necessary to modify the composition of the existing sensor membrane to reduce its response to anions (especially iodide and nitrate ions, which are commonly contained in seafood or vegetables).

Our previous research compared three positively charged sensor membranes, which have different charged states, containing oleylamine (OAm), trioctylmethylammonium chloride (TOMACl), and TDAB [29]. The research indicates that the ion selectivity of the sensor employing the incompletely dissociated lipid OAm is significantly lower than that of the other two sensors using the completely dissociated lipids TOMACl or TDAB. These findings facilitate the development of ion sensor membranes with different ion selectivity and provide new insights into creating a bitterness sensor that minimizes ion interference.

Therefore, this study aims to develop a bitterness sensor that is less affected by interfering anions, such as NO_3_^−^ and I^−^, by utilizing a partially dissociated substance as lipid substitutes for TDAB. According to our previous research, we selected three dissociated lipids for comparative studies: primary aliphatic amine OAm, secondary aliphatic amine dioctadecylamine (DODA), and tertiary aliphatic amine tridodecylamine (TDA).

## 2. Experiment

### 2.1. Reagents

The lipid polymer membrane is composed of lipid, a plasticizer, and support material. In this study, OAm, DODA, TDA, and TDAB were used as positively charged lipids in the membranes. OAm, DODA, and TDA were purchased from Tokyo Chemical Industry Co., Ltd., Tokyo, Japan. TDAB was purchased from Sigma-Aldrich Japan G.K. (Tokyo, Japan). The plasticizer used was NPOE purchased from Dojindo Laboratory Co. (Kumamoto, Japan), and the supporting material was PVC purchased from Fujifilm Wako Pure Chemicals Corporation (Osaka, Japan). Tetrahydrofuran (THF) purchased from Sigma-Aldrich Japan G.K. (Tokyo, Japan), was used as the organic solvent for the mixing of membrane components. The structures of these four lipids are shown in Figure 1.

To test the characteristics of ion selectivity for different anions, potassium chloride (KCl) and potassium iodide (KI) purchased from Fujifilm Wako Pure Chemicals Co. (Osaka, Japan) and potassium nitrate (KNO_3_) purchased from Hayashi Pure Chemical Industry Co. (Osaka, Japan) were used in anion solutions. To check the selectivity and sensitivity of the sensors towards taste substances, monosodium glutamate (MSG), tartaric acid, quinine hydrochloride, and tannic acid purchased from Kanto Chemical Co. (Tokyo, Japan) were used for taste samples.

A buffer solution containing 10 mM KCl was used to prepare all taste samples unless otherwise specified. For cleaning the membrane before subsequent measurements, a solution with 30 vol% ethanol was utilized. All reagents are of analytical grade and were used as received without further purification.

### 2.2. Fabrication of Lipid Polymer Membranes and Sensor Electrodes

Lipids (OAm, DODA, TDA, and TDAB), 1 mL of NPOE, and 800 mg of PVC, are dissolved in THF. The mixture is then poured into 90 mm Petri dishes to let THF evaporate and then form membranes. The membrane has a thickness of approximately 0.35 mm. After the membranes are formed on the Petri dishes, they are cut into pieces of same size and affixed to a hollow sensor probe, as is shown in Figure 2.

After placing the membranes onto the sensor probes, the probes are immersed into a reference solution (10 mM KCl water solution) for 24 h. Finally, a Ag wire coated with AgCl is injected into the probe after injecting an internal aqueous solution (3.33 M KCl, saturated with AgCl).

### 2.3. Apparatus

The measurement method adopts a two-electrode system, consisting of sensor electrodes and a reference electrode. A taste sensing system (TS-5000Z, from Intelligent Sensor Technology, Inc., Kanagawa, Japan) is utilized for the automatic measurement of the potential difference between sensor electrodes and reference electrode in various solutions. Figure 2 illustrates the structure of the detection unit and the appearance of the sensor electrode. Both the sensor electrode and the reference electrode are equipped with a Ag wire coated with a AgCl layer and are filled with an inner water solution containing KCl at a concentration of 3.33 M and saturated AgCl.

### 2.4. Procedure of Measurement

Initially, the sensor electrodes and the reference electrode are immersed in a reference solution (10 mM KCl) for 30 s to obtain the reference potential (*V*r). Subsequently, they are immersed in the sample solution for 30 s to obtain the sample potential (*V*s). The relative potential in the sample solution is calculated by taking the difference between *V*s and *V*r, and we usually take this relative potential as one of the sensor outputs, called the *relative value*. After that, the electrodes are immersed in another reference solution for light cleaning. During this process, some substances, especially highly hydrophilic substances, fall off the membrane, while high hydrophobic substances remain attached to the membrane. Subsequently, the sensor electrodes and the reference electrode are immersed in a reference solution for 30 s to obtain a new reference potential (*V*r’). The difference between *V*r’ and *V*r, which reflects the change in membrane potential caused by adsorption, is called the *CPA value* (*V*r’−*V*r). Finally, the sensor electrodes and the reference electrode are cleaned with a cleaning solution of 30 vol% EtOH in preparation for the next measurement cycle.

To ensure the reliability of the experimental data, we set the taste sensing system to perform the above measurement cycle five times and selected the last three times for analysis.

### 2.5. Experiment Configuration

#### 2.5.1. Response Test to KNO_3_ and KI of Various Concentrations

The purpose of this experiment is to determine whether the sensor electrodes with the OAm, DODA, and TDA membranes show a lower response to NO_3_^−^ and I^−^ at different concentrations than the TDAB membrane. The lipid concentration is set at 0.4 mM, consistent with the concentration of TDAB in C00. Based on the concentrations of I^−^ and NO_3_^−^ in food mentioned above, the concentrations of KNO_3_ are set at 1, 3, 10, and 30 mM, and the concentrations of KI are set at 0.1, 0.3, 1, and 3 mM. The solvent used is pure water. The experiment was conducted in three rounds, with four electrodes made for each lipid membrane, resulting in twelve measurements for each detection data point, and the averages and errors were recorded.

#### 2.5.2. Membrane Lipid Concentration-Dependent Test

The purpose of this experiment is to determine whether OAm, DODA, and TDA can function similarly to TDAB for detecting acidic bitter substances. The concentrations of these lipids in the membranes are set between 0.4 and 400 mM. This concentration range allows for the assessment of how varying lipid concentrations in the membranes influence sensor sensitivity towards IAAs, providing insights into the optimal lipid concentration for effective bitterness measurement. The sample is set as a standard acidic bitterness sample, with 0.01 vol% IAAs dissolved in 10 mM KCl solution. For each lipid and each concentration of the membrane, four electrodes were made, and three rounds of experiments were conducted, resulting in twelve measurements for each data point, with the average and error processed accordingly.

#### 2.5.3. Sensor Sensitivity Test for IAAs

The purpose of this experiment is to check the linearity of the sensor’s response and its sensitivity to IAAs using different lipid membranes. The lipid concentration is set at 0.4 mM. In this experiment, the concentrations of the IAAs samples are set at 0.0003, 0.001, 0.003, 0.01, and 0.03 vol%. For each lipid membrane, four electrodes were prepared, and three rounds of effective tests were conducted, meaning that each data point is the average of 12 measurements with error processing.

#### 2.5.4. Taste Selectivity Test

The purpose of this experiment is to test the response characteristics of the OAm, DODA, and TDA membranes to basic flavors in order to evaluate the strengths of their selectivity. All sensor membranes are prepared using the lipid of 0.4 mM to assess their responses to standard taste samples with the compositions and concentrations in Table 1. All samples are prepared with a solvent of 10 mM KCl solution. For each lipid and each concentration of the membrane, four electrodes were made, and three rounds of experiments were conducted, resulting in twelve measurements for each data point, with the average and error processed.

Typical sugars such as glucose or sucrose are non-electrolytes. These non-electrolytes will not affect the membrane’s potential, which means there will be no sensor output. Therefore, sweetness samples are not present in this experiment.

### 2.6. Performance Assessment

#### 2.6.1. Ion Selectivity Ratio

In this study, since we are mainly concerned with the influence of I^−^ and NO_3_^−^ on sensor responses, we defined the selectivity ratio *P* between these two ions as the index of ion selectivity (Equation (1)). According to the Hofmeister series, the response to I^−^ is generally higher than that to NO_3_^−^, so the *P* is usually larger than 1. The higher the *P* value is, the more sensitive the sensor’s response to ions is considered.
(1)$$P=\frac{\mathrm{Relative}\;\mathrm{value}\left(I^{-}\right)}{\mathrm{Relative}\;\mathrm{value}\left({\mathrm{NO}}_{3}^{-}\right)}$$

#### 2.6.2. Interference Resistance

We generally evaluate the sensor’s response to bitterness by comparing its response to a certain concentration of IAAs. However, their response to anions might also change along with the responses to bitterness, complicating the assessment of the sensor’s resistance to ionic interference. Therefore, to objectively compare the anti-interference ability of the bitterness sensor to specific anions, we define interference resistance (*IR*) as the ratio of the response of IAAs to the response of anions (Equation (2)). The parameter, *IR*, helps quantify the sensor’s ability to discriminate the bitterness response from ion interference, thereby providing a more objective measure of the sensor’s performance in selectively detecting bitterness in the presence of interfering ions.
(2)$$IR=\frac{\mathrm{Relative}\;\mathrm{value}\left(\mathrm{IAAs}\right)}{\mathrm{Relative}\;\mathrm{value}\left(\mathrm{anion}\right)}$$

The higher the *IR* value, the stronger the bitterness sensor’s ability to resist ion interference. It indicates that sensors with higher *IR* can interfere with lower concentrations of anions, making them more reliable for bitterness evaluation.

## 3. Results and Discussion

### 3.1. Evaluation of Ion Sensitivity

As is shown in Figure 3, relative values show a negative response. This is because the dissociation of lipids causes the membrane to carry a positive charge, and the interaction of anions with the membrane reduces the charge density near the membrane surface. We consider that greater changes in relative values indicate stronger responses. The statistical difference between relative values of the TDAB membrane and the amine membranes was determined using the *t*-test and is shown in Figure 3.

For all lipids, as the concentration of KNO_3_ and KI in the sample increases, the sensor response also increases. Among them, the electrodes using TDAB membranes exhibit the highest relative values to both NO_3_^−^ and I^−^ at highest lipid concentration, reaching −82.50 mV for 30 mM KNO_3_ and −115.36 mV for 3 mM KI. In contrast, OAm shows the lowest responses to NO_3_^−^ and I^−^, with responses of −43.03 mV and −74.03 mV, respectively. At all concentrations, the response of TDAB is higher than those of the other lipids, and the difference becomes more obvious as the sample concentration increases, which is supported by the *p* value of the *t*-test.

As for the relative values to KNO_3_, at 1 mM sample concentration, the response intensities of the various lipids are relatively close. However, TDAB exhibits a higher response compared to the other elements when concentration increases. OAm and DODA show lower response intensities, with minimal difference between them. On the other hand, TDA demonstrates a response slightly higher than those of OAm and DODA, but still lower than that of TDAB.

As for the relative values to KI, the response to KI is generally higher than that to KNO_3_, which is consistent with the predictions of the Hofmeister series, even when the sample concentration is lower than that of KNO_3_. We also observed that sensors with membranes using aliphatic amines (OAm, DODA, and TDA) exhibit lower and close responses than that with the ammonium salt (TDAB). The observation is consistent with the results of our previous study investigating the effects of cations on negative membranes [32], indicating that the membranes made of partially dissociated aliphatic amines have a lower surface charge density compared to those with ammonium salt lipid polymer membranes, which leads to weaker electrostatic interactions with anions [33].

### 3.2. Comparison of Ion Selectivity

By calculating the ion selectivity ratio *P*, we found that the ion selectivity ratios for the OAm, DODA, and TDA membranes are lower than TDAB at 0.4 mM lipid concentration (Table 2). A smaller *P* value indicates that the sensor has lower ion selectivity.

The *P* values for the membranes utilizing OAm (2.46 ± 0.08), DODA (2.90 ± 0.18), and TDA (2.29 ± 0.16) are all lower than those for the membranes utilizing TDAB (3.56 ± 0.64). It indicates that when partially dissociated lipids are used in place of fully dissociated lipids, the ion selectivity of the membrane electrodes for I^−^ and NO_3_^−^ is reduced. It is noteworthy that although the relative values of OAm and DODA (membrane electrodes) to various concentrations of KNO_3_ and KI are lower than those of TDA, their *P* values are higher than TDA’s *P* value.

We considered the possible reason for this phenomenon as follows. One study synthesized a host macromolecule for their research, which had a non-polar “cavity” and a “crown” structure composed of several trimethylammonium cations with electrical properties and strong polarity [34]. The study found that different anions in the Hofmeister series interact differently with the cavity and crown. For NO_3_^−^, the binding coefficient representing the binding strength of the crown to the cavity is about 1.89 times, while for I^−^, the binding coefficient of the crown is 3.44 times that of the cavity. In other words, in the interactions with the macromolecule, NO_3_^−^ is more likely to select the cavity structure compared to I^−^ at the binding sites. We infer that as it moves from OAm to DODA to TDA, the grade number (carbon chains number connected to the nitrogen atom) of amines increases, potentially making the structure near the membrane surface more complex and forming non-polar cavities composed of carbon chains. These cavities are more suitable for NO_3_^−^ entry, thereby increasing the response to NO_3_^−^. Therefore, the TDA membrane has a higher response to NO_3_^−^ than the OAm and DODA membranes. As for I^−^, it interacts more with the crown region than cavities, leading to a similar response from various amine membranes to I^−^; so towards amines, the responses are very close to each other. As a result, TDA has a relatively lower *P* value. However, the *P* value only reflects the performance of the sensor on ion selectivity. Our goal is to achieve a relatively lower response than bitterness, rather than strictly reducing ion selectivity. Thus, the resistance to interference of anions should be considered as well when evaluating the performance of the bitterness sensor.

### 3.3. Optimal Lipid Concentration for Bitterness Sensing

As shown in Figure 4a, the relative values of the OAm and TDAB membrane electrodes initially increases with lipid concentration, then decrease, reaching a peak at 4 mM. The DODA and TDA membrane electrodes also show higher relative values at low concentrations. The relative value of the TDAB membrane electrode suddenly drops from 40 mM and becomes almost zero at 400 mM. From the overall trend of relative values with varying concentrations, depending on whether using sensor membranes with fully dissociated or partially dissociated lipids, the relative values achieve their peak responses to IAAs at concentrations between 0.4 and 4 mM. Regarding the CPA values, they also show higher values at low concentrations and approach zero at high concentrations (above 40 mM). CPA values are crucial for both qualitative and quantitative assessments of bitterness, serving as an indicator of bitter aftertaste. Therefore, low-concentration lipid membranes are preferred as candidates for membranes of a bitterness sensor.

As mentioned in the measurement method, the bitterness response is due to the interaction between IAAs and the lipid polymer membranes, which involves both electrostatic and hydrophobic interactions. We hypothesize that as the charge density of the membrane increases, although the membrane’s capacity for electrostatic adsorption increases, its hydrophilicity also increases due to enhanced polarity, resulting in a decrease in hydrophobicity. The presence of the CPA values confirms the existence of hydrophobic binding between IAAs and the lipid polymer membranes. That the CPA values decrease with increasing lipid concentration indicates a reduction in the membrane’s hydrophobicity. Based on this, we concluded that lower lipid concentrations are more suitable for sensing hydrophobic bitter substances, such as IAAs. Therefore, we consistently use lipids at a concentration of 0.4 mM in subsequent experiments.

### 3.4. Evaluation on Interference Resistance

Here, we calculate the *IR* for nitrate and iodide ions using the ratio of the response to a standard bitter substance sample (0.01 vol% IAAs) to the response to 30 mM KNO_3_ and 1 mM KI.

As shown in Table 3, the *IR* of the OAm, DODA, and TDA membrane electrodes is improved compared to the TDAB membrane electrode at a lipid concentration of 0.4 mM. Among them, OAm showed the highest *IR* to NO_3_^−^ (*IR* = 2.28 ± 0.04), followed by TDA (*IR* = 2.06 ± 0.15), both significantly higher than TDAB. In terms of the *IR* to I^−^, TDA shows the highest *IR* with 2.23 ± 0.20, followed by OAm with 1.84 ± 0.04, both exceeding that of TDAB. These results indicate that membranes made with partially dissociated lipids show a higher resistance to ion interference compared to fully dissociated TDAB, especially for OAm and TDA.

### 3.5. Evaluation on Sensor Sensitivity

In Figure 5, the relative values show a linear relationship with the logarithm of the IAAs concentrations. Each relationship has been fitted with a response characteristic curve, modeled as *y* = *a*×log(*x*) + *b*, where “a” represents the increase in electrode response for every tenfold increase in the concentration of IAAs. This coefficient “a” can be considered an indicator of the sensor’s sensitivity to the bitter substance IAAs. As the coefficients of determination (*R*^2^) are all close to one, it can be concluded that OAm, DODA, and TDA all exhibit a good linear relationship with the logarithm of the IAAs concentrations. For every tenfold increase in IAAs concentration, the responses of the OAm, DODA, and TDA membrane electrodes increase by 59.01 mV, 44.77 mV, and 80.44 mV, respectively. The sensitivity of the TDA membrane electrode to IAAs, is even higher than that of the TDAB membrane electrode, which is 68.46 mV.

### 3.6. Evaluation on Taste Selectivity

As shown in Figure 6, the response of the TDAB membrane electrode, which is used as the lipid in the C00 sensor, obtained −116.94 mV to IAAs, which was significantly higher than other taste qualities. It exhibited relatively good selectivity for IAAs. Additionally, the CPA value to IAAs was −33.22 mV, indicating the degree of IAAs adsorption on the membrane and serving as an indicator of aftertaste in actual taste perception. Based on the measurement method of the TDAB membrane, we analyzed the taste selectivity of the OAm, DODA, and TDA membranes to comparatively evaluate their performance in bitterness selectivity.

As aliphatic amines, OAm, DODA, and TDA share similar properties. Like the TDAB membrane electrode, the OAm, DODA, and TDA electrodes also exhibited higher relative and CPA values for IAAs compared to other bitter substances. On the other hand, unlike the TDAB membrane electrode, the OAm, DODA, and TDA membrane electrodes showed relatively higher responses to acidic samples.

The reason is that OAm, DODA, and TDA are aliphatic amines, and higher concentrations of H^+^ promote their dissociation in neutral solution, thereby increasing the membrane’s own potential. To some extent, it also reflects that aliphatic amines, when used as membrane lipids, are indeed partially dissociated in reference solutions.

From the perspective of taste selectivity, partially dissociated amines are slightly inferior to TDAB due to their greater response to saltiness and umami. However, both TDAB and partially dissociated amines have their own advantages in taste selectivity; for instance, the TDAB membrane is not affected by sourness and umami, and the partially dissociated amine membranes are not affected by saltiness and astringency. Additionally, this does not significantly impact the effectiveness of partially dissociated lipids in reducing anion interference.

## 4. Conclusions

This study aims to develop an advanced bitterness sensor that reduces interference from common anions like nitrate (NO_3_^−^) and iodide (I^−^) found in seafood or vegetables. The central idea is to achieve this by incorporating partially dissociated amine OAm, DODA, or TDA into the sensor membrane instead of fully dissociated lipids.

Key findings from our study demonstrate that these amine-based membranes exhibit lower ion selectivity compared to the ammonium salt lipid TDAB, and they show enhanced resistance to interference of bitterness substances (IAAs in our study) from NO_3_^−^ and I^−^ ions. This is mainly due to the reduced charge density of the membrane surface brought about by partially dissociated lipids, which in turn mitigates the effects of the Hofmeister series. Among them, the TDA membrane shows the lowest ion selectivity, and higher interference resistance than TDAB.

An increasing trend of NO_3_^−^ responses as the grade number for amines increased was found, while I^−^ did not show it. NO_3_^−^ interacts more with “cavities” than with “crowns” which are formed on the surface of membranes, compared to I^−^. From OAm to DODA to TDA, the structure of surface membranes becomes more and more complex, therefore producing more cavities. Thus, the NO_3_^−^ response of amines showed a graded trend while I^−^ response were closer to each other.

The lipid concentration dependent experiment of the OAm, DODA, and TDA membranes revealed lower concentrations of these lipids are more suitable for bitterness sensing. Lower concentrations of lipids result in lower membrane polarity, which enhances hydrophobic interactions while retaining electrical interactions. This allows IAAs with strong hydrophobicity to interact more strongly with the membrane, maintaining a higher CPA value. Results showed that the TDA membrane has a higher IAAs relative value and CPA value than the TDA membrane at 0.4 mM.

From the results of sensor selectivity, we concluded that the TDA membrane’s sensitivity to IAAs was higher than the TDAB membrane, underscoring its superior performance and potential as a prime candidate for lipid of bitterness sensors.

The taste selectivity tests revealed that the OAm, DODA, and TDA membranes show a close taste selectivity due to their common substance characteristics. Although a considerable response to sourness and umami was found, their response to saltiness and astringency was lower than the TDAB membrane.

Considering the experimental data, including ion selectivity, interference resistance, sensitivity towards IAAs, and taste selectivity, TDA emerges as an optimal choice for further development in bitterness sensor technology. Future research will focus on enhancing the robustness of TDA-based sensors, exploring broader applications in food quality assessment, and extending to other industries where accurate taste evaluation is crucial.

## Figures and Tables

**Figure 1 sensors-24-05553-f001:**
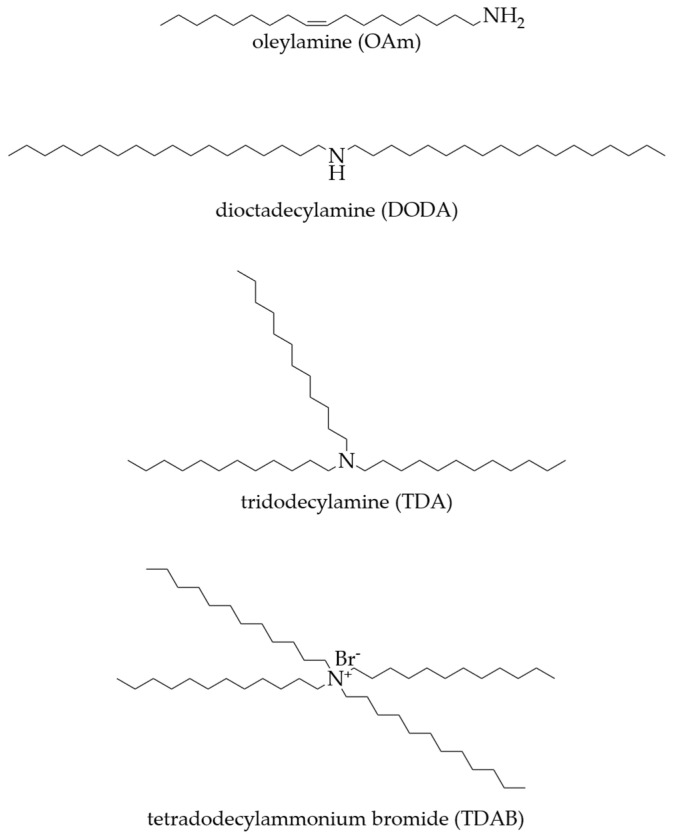
Structure of lipids utilized in sensor membranes: OAm, DODA, TDA, and TDAB.

**Figure 2 sensors-24-05553-f002:**
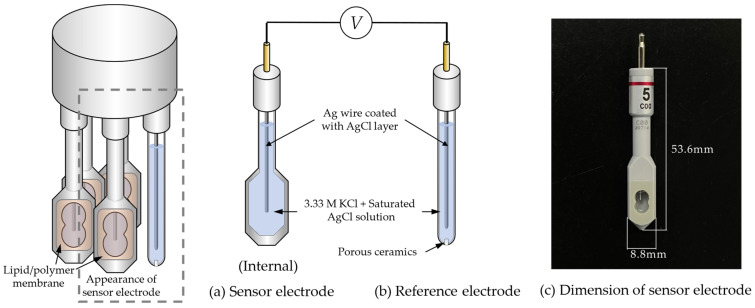
The structure of detection unit.

**Figure 3 sensors-24-05553-f003:**
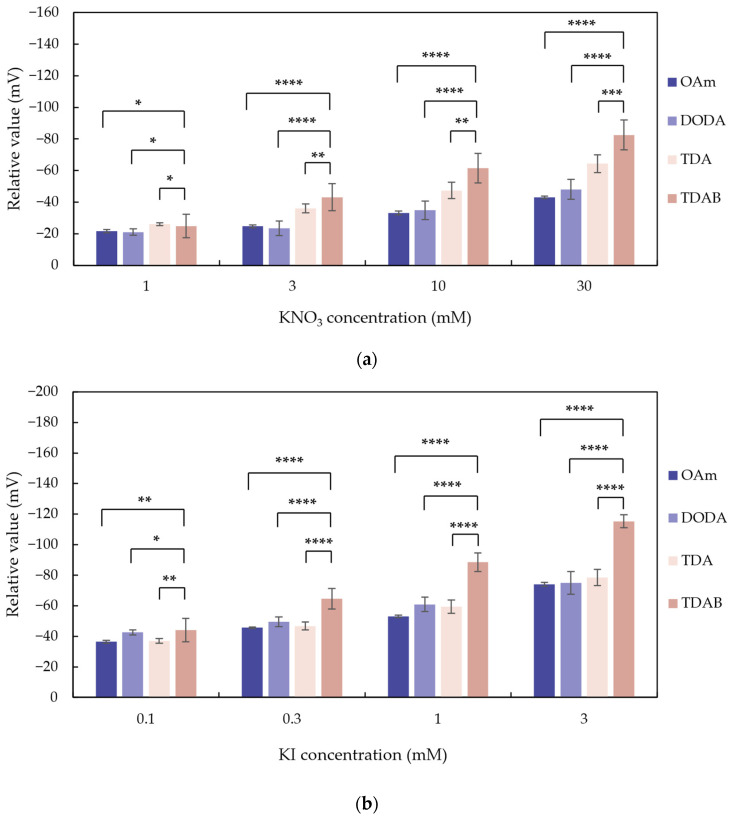
The concentration dependence of (**a**) KNO_3_ and (**b**) KI on membranes using different lipids (OAm, DODA, TDA, and TDAB) at 0.4 mM. The results are expressed as the means ± SD (*n* = 12). Asterisks represent statistical significance (* *p* < 0.05; ** *p* < 0.01; *** *p* < 0.001; **** *p* < 0.0001).

**Figure 4 sensors-24-05553-f004:**
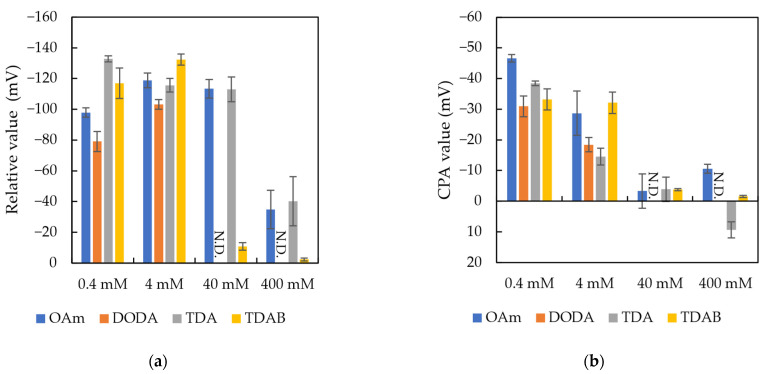
Results of (**a**) relative values and (**b**) CPA values towards IAAs for the OAm, DODA, TDA, and TDAB membranes with lipid concentrations at 0.4, 4, 40, and 400 mM. DODA is saturated at the concentrations of 40 mM and 400 mM and cannot form a uniform lipid polymer membrane. The results are expressed as the means ± SD (*n* = 12).

**Figure 5 sensors-24-05553-f005:**
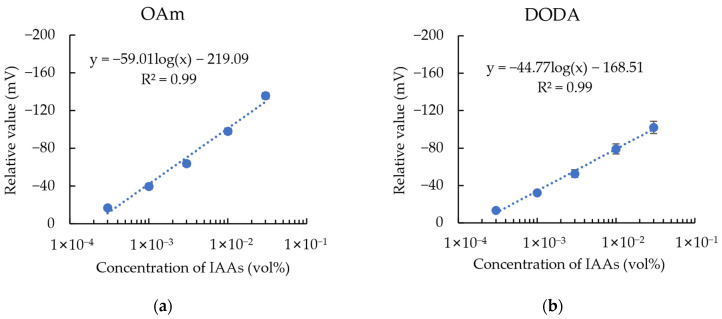
Concentration dependence on Iso-α acids (IAAs) using membrane electrodes with (**a**) OAm, (**b**) DODA, (**c**) TDA, and (**d**) TDAB. The results are expressed as the means ± SD (*n* = 12).

**Figure 6 sensors-24-05553-f006:**
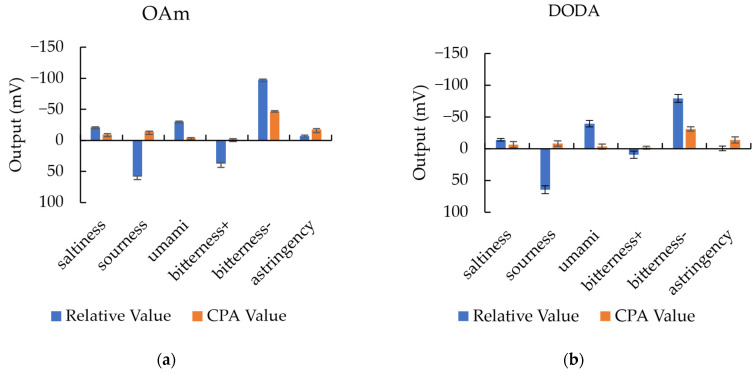
Responses for basic tastes using (**a**) OAm, (**b**) DODA, (**c**) TDA, and (**d**) TDAB membrane electrodes. The results are expressed as the means ± SD (*n* = 12).

**Table 1 sensors-24-05553-t001:** Compositions and concentrations of standard taste samples.

Taste Quality	Composition	Concentration
Sourness	Tartaric acid	3 mM
Umami	Monosodium glutamate	100 mM
Bitterness (+)	Quinine hydrochloride	0.1 mM
Bitterness (−)	Iso-α acids	0.01 vol%
Saltiness	Potassium chloride	300 mM
Astringency	Tannic acid	0.05 wt%

**Table 2 sensors-24-05553-t002:** *P* values of the OAm, DODA, TDA, and TDAB membrane electrodes to KI and KNO_3_ at 1 mM sample concentration.

Lipid	*P* (KI/KNO_3_ at 1 mM)
OAm	2.46 ± 0.08
DODA	2.90 ± 0.18
TDA	2.29 ± 0.16
TDAB	3.56 ± 0.64

**Table 3 sensors-24-05553-t003:** Interference Resistance (*IR*) of OAm, DODA, TDA, and TDAB (0.4 mM) membrane electrodes to NO_3_^−^ (30 mM) and I^−^ (1 mM).

Lipid	*IR* (NO_3_^−^)	*IR* (I^−^)
OAm	2.28 ± 0.04	1.84 ± 0.04
DODA	1.65 ± 0.25	1.30 ± 0.13
TDA	2.06 ± 0.15	2.23 ± 0.20
TDAB	1.42 ± 0.23	1.32 ± 0.16

## Data Availability

The data are available on request.

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
