# Peer review of "Development of a Bitterness Sensor Using Partially Dissociated Amine Compounds"

_sensors, 2024, doi:10.3390/s24175553_

Round 1

Reviewer 1 Report

Comments and Suggestions for Authors

This manuscript aimed to improve bitter taste sensors by experimentally testing multiple sensor materials with the goal of minimizing interference from common anions. In the introduction section, the problems with a conventional sensor and the characteristics of the sensors that the authors have developed were sufficiently explained, and the direction for improvement is presented. In the experiments, four lipids were tested. The experimental results showed that the membranes with partially dissociated amines significantly reduced anion interference. The authors further evaluated the sensitivity of lipids and concluded that one lipid had significantly higher sensitivity and was useful for detecting bitterness.

The manuscript fits within the scope of the journal. The reviewer would recommend it for acceptance after the minor points listed below are addressed.

Minor comment:
- In Figure 2, add the dimension of the electrode. Also, a photo of the electrode is useful for readers.

Author Response

Comments 1: In Figure 2, add the dimension of the electrode. Also, a photo of the electrode is useful for readers.

Response 1: Thank you very much for your suggestion. We have added the photo and dimension of the electrode in Figure 2, to describe it more clearly for readers.

The revised content can be found in line 150-151, 155-156 of the revised manuscript.

Reviewer 2 Report

Comments and Suggestions for Authors

The authors aimed to develop an advanced bitterness sensor designed to minimize interference from common anions such as nitrate (NO₃-) and iodide (I-). To address the issue that the conventional bitter sensor (C00) uses fully dissociated quaternary ammonium salt tetradecyl ammonium bromide (TDAB) and exhibits high responses to these anions, leading to inaccurate bitterness assessments, the investigators incorporated partially dissociated amine compounds into the sensor membrane. Multiple parameters were tested including ion selectivity, interference resistance to anion, sensitivity to iso-alpha acids, bitter compounds in beer. The results indicate that TDA emerges as an optimal choice for further development in bitterness sensor technology, which may be useful for food quality assessment. Being not an expert in physics, I have a few questions as a biologist.

1.      In the standard taste samples, tastants for four of the five basic taste qualities were included but sweet taste was not tested. Would it be good to include it as a control? If it is not applicable, would it be good to discuss it briefly?   

2.      The outputs of multiple measurements (e.g., ion sensitivity) are negative values. Would it be good to make the bars downward (Figure 1, 4, 5, 6) to represent the negative values?

3.      Are there appropriate statistical analyses for comparisons of data among groups and within groups in Figure 3,  Table 2, Table 3?

Comments on the Quality of English Language

 In line 262, what is the meaning of “It.”?

  In line 269, did the authors mean to say Comparison instead of Comparasion?

Author Response

Comments 1: In the standard taste samples, tastants for four of the five basic taste qualities were included but sweet taste was not tested. Would it be good to include it as a control? If it is not applicable, would it be good to discuss it briefly?

Response 1: Thank you for pointing that out. Exactly, adding a sweetness sample would make the taste selectivity of the sensor more convincing. In practice, however, typical sugars such as glucose or sucrose, are non-electrolytes. These non-electrolytes will not affect the membrane potential, which means there will be no sensor output. Therefore, the lack of sweetness samples does not theoretically affect taste selectivity.

To dispel readers' doubts, we have also added the following content in lines 214 to 216 of the revised manuscript:

‘Typical sugars such as glucose or sucrose, are non-electrolytes. These non-electrolytes will not affect the membrane potential, which means there will be no sensor output. Therefore, the lack of sweetness samples does not theoretically affect taste selectivity, so sweetness samples were not set in this experiment.’

Comments 2: The outputs of multiple measurements (e.g., ion sensitivity) are negative values. Would it be good to make the bars downward (Figure 1, 4, 5, 6) to represent the negative values?

Response 2: Thank you for your advice.

In the original scenario, we made bars downward to represent negative data. In latter version, to represent and compare numerical sizes more intuitively, we changed the bars upward to represent negative values, because readers can see the changing trend and compare different data more easily. For positively charged sensor membrane, almost all outputs are negative. In fact, the intensity of taste is related to the absolute value of the output, but not to the positive or negative value of the output. We consider that this way is more intuitive for readers who are new to understanding this membrane structure.

Comments 3: Are there appropriate statistical analyses for comparisons of data among groups and within groups in Figure 3, Table 2, Table 3?

Response 3: Your good advice is very much appreciated.

For Figure 3, we additionally calculated the significance of the differences between the measurement results of the sample groups using t-test. The results obtained from t-test are consistent with the previous conclusions. The relevant content has been added in lines 247, 251-253, 260-261 of the manuscript.

For Table 2 and Table 3, errors are not included in previous calculated values. We have added calculated values with errors. Revised content can be found in line 283-284 and 348-350.

Comments 4: In line 262, what is the meaning of “It.”?

Response 4: Thank you for pointing that out. That is an editing error. We have already deleted it in line 262 (line 270 in the new manuscript).

Comments 5: In line 269, did the authors mean to say Comparison instead of Comparasion?

Response 5: Yes, it should be 'Comparison'. We have already corrected it in line 269 (line 277 in the new manuscript).
